# An Asymptotically Optimal Batched Algorithm for the Dueling Bandit Problem

**Arpit Agarwal**
Columbia University
arpit.agarwal@columbia.edu

**Rohan Ghuge**
University of Michigan
rghuge@umich.edu

**Viswanath Nagarajan**
University of Michigan
viswa@umich.edu

## Abstract

We study the $K$-armed dueling bandit problem, a variation of the traditional multi-armed bandit problem in which feedback is obtained in the form of pairwise comparisons. Previous learning algorithms have focused on the *fully adaptive* setting, where the algorithm can make updates after every comparison. The "batched" dueling bandit problem is motivated by large-scale applications like web search ranking and recommendation systems, where performing sequential updates may be infeasible. In this work, we ask: *is there a solution using only a few adaptive rounds that matches the asymptotic regret bounds of the best sequential algorithms for $K$-armed dueling bandits?* We answer this in the affirmative *under the Condorcet condition*, a standard setting of the $K$-armed dueling bandit problem. We obtain asymptotic regret of $O(K^2 \log^2(K)) + O(K \log(T))$ in $O(\log(T))$ rounds, where $T$ is the time horizon. Our regret bounds nearly match the best regret bounds known in the fully sequential setting under the Condorcet condition. Finally, in computational experiments over a variety of real-world datasets, we observe that our algorithm using $O(\log(T))$ rounds achieves almost the same performance as fully sequential algorithms (that use $T$ rounds).

## 1 Introduction

The *$K$-armed dueling bandit* problem is a variation of the traditional multi-armed bandit problem in which feedback is obtained in the form of pairwise preferences. This problem has applications in a wide-variety of domains like search ranking, recommendation systems and sports ranking where eliciting qualitative feedback is easy while real-valued feedback is not easily interpretable; thus, it has been a popular topic of research in the machine learning community (see, for example, [51, 49, 47, 5, 54, 52, 53, 21, 32, 35, 36, 42, 18]).

Previous learning algorithms have focused on a *fully adaptive* setting; that is, the learning algorithm can make updates in a sequential fashion. Such updates might be impractical in large systems; for example, consider web-search ranking where the goal is to provide a list (usually *ranked*) of candidate documents to the user of the system in response to a query [41, 33, 50, 31]. Modern day search engines use hundred of parameters to compute a ranked list in response to a query, and online learning frameworks (based on user feedback) have been invaluable in automatically tuning these parameters [38]. However, given the scale of the system, it may be infeasible to adapt after each interaction: users may make multiple queries in a short time or multiple users may simultaneously query the system. Hence, we prefer solutions with *limited rounds* of adaptivity. Concretely, we ask: *is there a solution using only a few adaptive rounds that matches the asymptotic regret bounds of the best sequential algorithms for $K$-armed dueling bandits?*

This "batched" dueling bandit problem was introduced recently in [2]. Here, the learning algorithm's actions are partitioned into a limited number of rounds. In each round/batch, the algorithm commits to a *fixed set* of pairwise comparisons, and the feedback for all these comparisons is received

simultaneously. Then, the algorithm uses the feedback from the current batch of comparisons to choose comparisons for the next batch. [2] studied this problem under two different conditions: (i) the strong stochastic transitivity and stochastic triangle inequality (SST+STI) condition, which enforces a certain linear ordering over the arms; (ii) the Condorcet condition, which requires one arm to be superior to all others. Under SST+STI, their work provided almost tight upper and lower bounds on the trade-off between number of rounds and regret; in particular, they showed that one can achieve worst-case regret of $O(K \log^2 T)$ using $\Theta(\log T)$ rounds ($T$ is the time-horizon).[1] Under the Condorcet condition, which is more general than SST+STI, they achieved a regret upper bound of $O(K^2 \log T)$ in $O(\log T)$ rounds. Previous work [54, 35] on fully sequential algorithms has shown that it is possible to achieve an *asymptotic* upper bound of $O(K^2 + K \log T)$ under the Condorcet condition. Very recently, [43] improved the sequential regret bound even further by obtaining regret $O(K \log T)$, which is the best possible even in the special case of SST+STI [49]. In the batched setting, the upper bound of [2] does not achieve this asymptotic optimality, irrespective of the number of batches, due to the presence of a $K^2$ *multiplicative* factor in the regret bound. Their work left open the possibility of obtaining a batched algorithm achieving asymptotic optimality under the Condorcet condition. In this paper, we nearly resolve this question, by providing an algorithm with $O(K^2 \log^2 K + K \log T)$ regret in $\Theta(\log T)$ rounds, under the Condorcet condition.

## 1.1 Contributions

- We design an algorithm, denoted C2B, for the batched dueling bandit problem, and analyze its regret under the Condorcet condition. This algorithm achieves a smooth trade-off between the expected regret and the number of batches, $B$.

- Crucially, when $B = \log(T)$, our regret bounds nearly match the best regret bounds [35, 54] known in the fully sequential setting. Hence, our results show that $O(\log T)$ rounds are sufficient to achieve asymptotically optimal regret as a function of $T$.

- Our results rely on new ideas for showing that the Condorcet winner arm can be 'trapped' using few adaptive rounds with high (constant) probability while incurring a reasonable amount of regret. We can then integrate over this space of probabilities to obtain a bound on the expected regret (in the same vein as [54]). Once the Condorcet arm is 'trapped', we can quickly eliminate all other 'sub-optimal' arms and minimize regret in the process.

- Finally, we run computational experiments to validate our theoretical results. We show that C2B, using $O(\log T)$ batches, achieves almost the same performance as fully sequential algorithms (which effectively use $T$ batches) over a variety of real datasets.

## 1.2 Preliminaries

The $K$-*armed dueling bandit* problem [49] is an online optimization problem, where the goal is to find the best among $K$ bandits $\mathcal{B} = \{1, \ldots, K\}$ using noisy pairwise comparisons with low *regret*. In each time-step, a noisy comparison between two arms (possibly the same), say $(i, j)$, is performed. The outcome of the comparison is an independent random variable, and the probability of picking $i$ over $j$ is denoted $p_{i,j} = \frac{1}{2} + \Delta_{i,j}$ where $\Delta_{i,j} \in (-\frac{1}{2}, \frac{1}{2})$. Here, $\Delta_{i,j}$ can be thought of as a measure of distinguishability between the two arms, and we use $i \succ j$ when $\Delta_{i,j} > 0$. We also refer to $\Delta_{i,j}$ as the *gap* between $i$ and $j$.

This problem has been studied under various conditions on the pairwise probabilities $p_{i,j}$'s. One such condition is the *strong stochastic transitivity* and *stochastic triangle inequality* (SST+STI) condition where there exists an ordering over arms, denoted by $\succeq$, such that for every triple $i \succeq j \succeq k$, we have $\Delta_{i,k} \geq \max\{\Delta_{i,j}, \Delta_{j,k}\}$, and $\Delta_{i,k} \leq \Delta_{i,j} + \Delta_{j,k}$ [49, 51]. In this paper, we work under the well-studied Condorcet winner condition, which is much more general than the SST+STI condition [47, 54, 35]. We say that arm $i$ is a *Condorcet winner* if, and only if, $p_{i,j} > \frac{1}{2}$ for all $j \in \mathcal{B} \setminus \{i\}$. The *Condorcet condition* means that there exists a Condorcet winner.

Throughout the paper, we let $a^*$ refer to the Condorcet arm. To further simplify notation, we define $\Delta_j = \Delta_{a^*, j}$; that is, the gap between $a^*$ and $j$. We define the *regret* per time-step as follows: suppose arms $i_t$ and $j_t$ are chosen in time-step $t$, then the regret $r(t) = \frac{\Delta_{i_t} + \Delta_{j_t}}{2}$. The cumulative regret up to time $T$ is $R(T) = \sum_{t=1}^{T} r(t)$, where $T$ is the time horizon, and it's assumed that $K \leq T$. The

---

[1]They also gave a more complicated algorithm with regret $O(K \log^2 K \log T)$ in $O(\log T + \log K \log \log K)$ rounds, under the SST+STI condition.

Table 1: An overview of our results. Note that $\widetilde{O}$ hides polylog$(K)$ factors.

| Setting | Regret |
|---|---|
| Fully Adaptive | $O\left(\frac{K^2}{\Delta_{\min}^2}\right) + \sum_{j:\Delta_j>0} O\left(\frac{\log T}{\Delta_j}\right)$ [54, 35] |
| $T$ rounds | $\sum_{j:\Delta_j>0} O\left(\frac{\log(T)}{\Delta_j}\right)$ [43] |
| Batched | |
| | $\sum_{j:\Delta_j>0} O\left(KT^{1/B} \cdot \frac{\log(T)}{\Delta_j}\right)$ [2] |
| $B$ rounds | |
| **Our Algorithm** | |
| $B$ rounds | $\widetilde{O}\left(T^{\frac{1}{B}} \cdot \frac{K^2}{\Delta_{\min}^2}\right) + O\left(T^{\frac{2}{B}} \cdot K^2\right) + \sum_j O\left(T^{\frac{1}{B}} \cdot \frac{\log(T)}{\Delta_j}\right)$ |
| $\log(T)$ rounds | $\widetilde{O}\left(\frac{K^2}{\Delta_{\min}^2}\right) + \sum_j O\left(\frac{\log(T)}{\Delta_j}\right)$ |

cumulative regret can be equivalently stated as $R(T) = \frac{1}{2}\sum_{j=1}^{K} T_j \Delta_j$, where $T_j$ denotes the number comparisons involving arm $j$. The goal of an algorithm is to minimize the cumulative $R(T)$. We define $\Delta_{\min} = \min_{j:\Delta_j>0} \Delta_j$ to be the smallest non-zero gap of any arm with $a^*$.

## 1.3 Batch Policies

In traditional bandit settings, actions are performed *sequentially*, utilizing the results of *all prior actions* in determining the next action. In the batched setting, the algorithm must commit to a round (or *batch*) of actions to be performed *in parallel*, and can only observe the results after all actions in the batch have been performed. More formally, given a number $B$ of batches, the algorithm proceeds as follows. In each batch $r = 1, 2, \ldots B$, the algorithm first decides on the comparisons to be performed; then, *all* outcomes of the batch-$r$ comparisons are received simultaneously. The algorithm can then, *adaptively*, select the next batch of comparisons. Note that even the size of the next batch can be adaptively decided based on the observations in previous batches. Finally, the total number of comparisons (across all batches) must sum to $T$. We assume that the values of $T$ and $B$ are known. Observe that when $T = B$, we recover the fully sequential setting.

## 1.4 Results and Techniques

We provide a overview of our results and prior results in Table 1. Given any integer $B \geq 1$, we obtain a $B$-round algorithm for the dueling bandit problem. We provide both high-probability and expected regret bounds, stated in the following theorems.

**Theorem 1.1.** *For any integer $B \geq 1$, there is an algorithm for the $K$-armed dueling bandit problem that uses at most $B$ rounds with the following guarantee. For any $\delta > 0$, with probability at least $1 - \delta - \frac{1}{T}$, its regret under the Condorcet condition is at most*

$$R(T) \leq O\left(T^{1/B} \cdot \frac{K^2 \log(K)}{\Delta_{\min}^2} \cdot \log\left(\frac{\log K}{\Delta_{\min}}\right)\right) + O\left(T^{2/B} \cdot K^2 \cdot \sqrt{\frac{1}{\delta}}\right)$$
$$+ \sum_{j \neq a^*} O\left(\frac{T^{1/B} \cdot \log(KT)}{\Delta_j}\right).$$

**Theorem 1.2.** *For any integer $B \geq 1$, there is an algorithm for the $K$-armed dueling bandit problem that uses at most $B$ rounds, with expected regret under the Condorcet condition at most*

$$\mathbb{E}[R(T)] = O\left(T^{1/B} \cdot \frac{K^2 \log(K)}{\Delta_{\min}^2} \cdot \log\left(\frac{\log K}{\Delta_{\min}}\right)\right) + O\left(T^{2/B} \cdot K^2\right)$$
$$+ \sum_{j \neq a^*} O\left(\frac{T^{1/B} \cdot \log(KT)}{\Delta_j}\right).$$

When the number of rounds $B = \log(T)$, we obtain a batched algorithm that achieves the asymptotic optimality (in terms of $T$), even for sequential algorithms. We formalize this observation in the following corollary.

**Corollary 1.3.** *There is an algorithm for the $K$-armed dueling bandit problem that uses at most $\log(T)$ rounds, with expected regret under the Condorcet condition at most*

$$\mathbb{E}[R(T)] = O\left( \frac{K^2 \log(K)}{\Delta_{\min}^2} \cdot \log\left( \frac{\log K}{\Delta_{\min}} \right) \right) + \sum_{j \neq a^*} O\left( \frac{\log(KT)}{\Delta_j} \right).$$

By a lower-bound result from [2], it follows that no algorithm can achieve $O(\frac{K}{\Delta_{\min}} \cdot poly(\log T))$ regret using $o(\frac{\log T}{\log \log T})$ rounds, even under the SST+STI condition. So, the $O(\log T)$ rounds required to achieve asymptotic optimality in Corollary 1.3 is nearly the best possible.

**Technical Challenges.** The only prior approach for batched dueling bandits (under the Condorcet condition) is the algorithm PCOMP from [2], which performs all-pairs comparisons among arms in an active set. Such an approach cannot achieve regret better than $O(K^2 \log T)$ because the active set may remain large throughout. In order to achieve better regret bounds, [2] focus on the stronger SST+STI condition. In this setting, their main idea is to first sample a *seed set*, and use this seed set to eliminate sub-optimal arms. Their algorithm proceeds by performing all pairwise comparisons between the seed set and the set of active arms. However, the analysis of these 'seeded comparison' algorithms crucially rely on the total-ordering imposed by the SST and STI assumptions. Unfortunately, there is no such structure to exploit in the Condorcet setting: if the seed set does not contain the Condorcet winner, we immediately incur high regret.

The existing fully sequential algorithms such as RUCB [54] and RMED [35] are *highly adaptive* in nature. For instance, RUCB plays each *candidate* arm against an optimistic *competitor* arm using upper confidence bounds (UCB) on pairwise probabilities. This allows RUCB to quickly filter out candidates and uncover the Condorcet arm. Similarly, RMED plays each arm against a carefully selected *competitor* arm that is likely to beat this arm. However, such *competitors* can change frequently over trials in both RUCB and RMED. Since the batched setting requires comparisons to be predetermined, we do not have the flexibility to adapt to such changes in *competitors*. Hence, these existing fully sequential algorithms cannot be easily implemented in our setting.

Furthermore, we might also be tempted to consider an *explore-then-exploit* strategy where we first *explore* to find the Condorcet arm and *exploit* by playing this arm for remaining trials. However, this strategy is likely to fail because identifying the Condorcet arm with high probability might involve performing many comparisons, directly leading to high $(\Omega(K^2 \log T))$ regret; on the other hand, if the Condorcet winner is not identified with high probability, the exploit phase becomes expensive. This motivated us to consider algorithms that allow some form of *recourse*; that is, unless an arm is found to be sub-optimal, it must be given the opportunity to participate in the comparisons (as it could be the Condorcet winner).

The idea behind our algorithm is to identify the Condorcet winner $a^*$ in a small *expected* number of rounds, after which it uses this arm as an "anchor" to eliminate sub-optimal arms while incurring low regret. To identify the best arm, in each round we define a candidate arm and compare it against arms that it "defeats". Arms that are not defeated by the candidate arm are compared to *all* active arms: this step ensures that the Condorcet winner is eventually discovered. We show that $a^*$ becomes the candidate, and *defeats all other arms* within a small number of rounds (though the algorithm may not know if this has occurred). Additionally, once this condition is established, it remains invariant in future rounds. This allows us to eliminate sub-optimal arms and achieve low regret.

**Comparison to** RUCB**.** Initially, RUCB puts all arms in a pool of potential champions, and "optimistically" (using a upper confidence bound) performs all pairwise comparisons. Using these, it constructs a set of candidates $C$. If $|C| = 1$, then that arm is the hypothesised Condorcet winner and placed in a set $B$. Then, a randomized strategy is employed to choose a champion arm $a_c$ (from sets $C$ and $B$) which is compared to arm $a_d$ which is most likely to beat it. The pair $(a_c, a_d)$ is compared, the probabilities are updated and the algorithm continues. Although our algorithm also seeks to identify the best arm, we do not employ the UCB approach nor do we use any randomness. In their analysis, [54] show that the best arm eventually enters the set $B$, and remains in $B$: we also show a similar property for our algorithm in the analysis. Finally, similar to the analysis of [54], we first give a

high-probability regret bound for our algorithm which we then convert to a bound on the expected regret.

## 2 Related Work

The $K$-armed dueling bandit problem has been widely studied in recent years (we refer the reader to [46] for a comprehensive survey). Here, we survey the works that are most closely related to our setting. This problem was first studied in [49] under the SST and STI setting. The authors obtained a worst-case regret upper bound of $\widetilde{O}(K \log T / \Delta_{\min})$ and provided a matching lower bound. [51] considered a slightly more general version of the SST and STI setting and achieved an instance-wise optimal regret upper bound of $\sum_{j:\Delta_j>0} O\left(\log(T)/\Delta_j\right)$. Since, the SST+STI condition imposes a total order over the arms and might not hold for real-world datasets, [47] initiated the study of dueling bandits under the Condorcet winner condition. [47] proved a $O(K^2 \log T/\Delta_{\min})$ regret upper bound under the Condorcet condition, which was improved by [54] to $O(K^2/\Delta_{\min}^2) + \sum_{j:\Delta_j>0} O(\log T/\Delta_j^2)$. [35] achieved a similar but tighter KL divergence-based bound, which is shown to be *asymptotically instance-wise optimal* (even in terms constant factors). There are also other works that improve the dependence on $K$ in the upper bound, but suffer a worse dependence on $\Delta_j$'s [53]. This problem has also been studied under other noise models such as utility based models [5] and other notions of regret [18]. Alternate notions of winners such as Borda winner [32], Copeland winner [52, 36, 48], and von Neumann winner [21] have also been considered. There are also several works on extensions of dueling bandits that allow multiple arms to be compared at once [45, 3, 44].

All of the aforementioned works on the dueling bandits problem are limited to the sequential setting. Recently, [2] initiated the study of the batched version of the $K$-armed dueling bandits. Their main results are under the SST and STI setting. They give two algorithms, called SCOMP and SCOMP2, for the batched $K$-armed dueling bandit problem. For any integer $B$, SCOMP uses at most $B+1$ batches and has expected regret bounded by $\sum_{j:\Delta_j>0} O(\sqrt{K}T^{1/B} \log(T)/\Delta_j)$. When $B = \log(T)$, this nearly matches (up to a factor of $\sqrt{K}$) the best known instance-dependent regret bound of $\sum_{j:\Delta_j>0} O(\log(T)/\Delta_j)$ obtained by [49]. SCOMP2 aims to achieve better worst-case regret: it uses at most $2B+1$ batches, and has regret $O\left(KBT^{1/B} \log(T)/\Delta_{\min}\right)$. Thus, when $B = \log(T)$, the expected worst-case regret is $O\left(K\log^2(T)/\Delta_{\min}\right)$, matching the best known result in the sequential setting up to an additional logarithmic factor. Under the Condorcet condition, [2] give a straightforward pairwise comparison algorithm (PCOMP), that achieves expected regret bounded by $O(K^2 \log(T)/\Delta_{\min})$ in $\log(T)$ batches. They also provide a nearly matching lower bound of $\Omega(\frac{KT^{1/B}}{B^2 \Delta_{\min}})$ for any $B$-batched algorithm. This implies that our bound (for $B$-round algorithms) in Theorem 1.2 cannot be significantly improved.

Recently, [43] designed a fully adaptive algorithm achieving an optimal regret of $\sum_{j:\Delta_j>0} \frac{O(\log T)}{\Delta_j}$ for dueling bandits under the Condorcet setting. This algorithm is based on the idea of *dueling* two classical bandit (MAB) algorithms against each other in a repeated zero-sum game with carefully designed rewards. The reward for one algorithm depends on the actions of the other; hence, these algorithms need to achieve *best-of-both-worlds* guarantee for both stochastic and adversarial settings. However, the approach of [43] is not directly applicable to the *batched* setting that we consider. This is because, as shown by [23], any $B$-round algorithm for batched MAB in the adversarial setting has regret $\Omega(T/B)$.

There has also been substantial work on best-arm or top-$k$ identification using pairwise comparisons with limited adaptivity. [15, 14, 19] considered this problem under the noisy pairwise comparison setting, which is a special case of SST+STI. They showed that constant number of rounds of adaptivity are sufficient to solve these problem with the optimal sample complexity. [4] showed that one can also solve this problem under SST in constant number of rounds with the optimal sample complexity. However, these existing results focus on the SST setting, whereas we focus on the more general Condorcet winner setting. Moreover, these existing works focus on sample complexity for best-arm identification whereas our goal is regret minimization.

## 3 The Batched Algorithm

In this section, we describe a $B$-round algorithm for the $K$-armed dueling bandit problem under the Condorcet condition. Recall that given a set of $K$ arms, $\mathcal{B} = \{1, \dots, K\}$, and a positive integer

$B \leq \log(T)$, we wish to find a sequence of $B$ batches of noisy comparisons with low regret. Given arms $i$ and $j$, recall that $p_{i,j} = \frac{1}{2} + \Delta_{i,j}$ denotes the probability of $i$ winning over $j$ where $\Delta_{i,j} \in (-1/2, 1/2)$. We use $a^*$ to denote the Condorcet winner; recall that $a^*$ is a Condorcet winner if $p_{a^*,j} \geq 1/2$ for all $j \in \mathcal{B}$. To simplify notation, we use $\Delta_j = \Delta_{a^*,j}$. Before describing our algorithm, we first define some notation. We use $\mathcal{A}$ to denote the current set of *active* arms; i.e., the arms that have not been eliminated. We will use index $r$ for rounds or batches. If pair $(i, j)$ is compared in round $r$, it is compared $q_r = \lfloor q^r \rfloor$ times where $q = T^{1/B}$. We define the following quantities at the *end* of each round $r$:

- $N_{i,j}(r)$ is the total number of times the pair $(i, j)$ has been compared.
- $\widehat{p}_{i,j}(r)$ is the frequentist estimate of $p_{i,j}$, i.e.,

$$\widehat{p}_{i,j}(r) = \frac{\# \; i \text{ wins against } j \text{ until end of round } r}{N_{i,j}(r)} \,. \tag{1}$$

- Two confidence-interval radii for each $(i, j)$ pair:

$$c_{i,j}(r) = \sqrt{\frac{2 \log(2K^2 q_r)}{N_{i,j}(r)}} \qquad \text{and} \qquad \gamma_{i,j}(r) = \sqrt{\frac{\log(K^2 BT)}{2N_{i,j}(r)}} \tag{2}$$

We now describe our $B$-round algorithm, called CATCHING THE CONDORCET WINNER IN BATCHES (or, C2B). At a high-level, the algorithm identifies the best arm $a^*$ in a small expected number of rounds, after which it uses this arm as an "anchor" to eliminate sub-optimal arms while incurring low regret. In every round $r$, we do the following:

1. We define a *defeated set* $D_r(i)$ for every active arm $i$; this set comprises arms that are defeated *with confidence* by $i$. Specifically, $j \in D_r(i)$ if $\widehat{p}_{i,j}(r-1) > 1/2 + c_{i,j}(r-1)$.

2. Then, we define a *candidate* $i_r$ as the arm that defeats the most number of arms; that is, $i_r = \arg\max_{i \in \mathcal{A}} |D_r(i)|$.

3. For every arm $i \neq i_r$:
   - If $i \in D_r(i_r)$, then we compare $i$ to $i_r$ for $q_r$ times. The idea here is to use $i_r$ as an anchor against $i$. We will show that $a^*$ becomes the candidate $i_r$ in a small number of rounds. Then, this step ensures that we eliminate arms efficiently using $a^*$ as an anchor.
   - If $i \notin D_r(i_r)$, then $i$ is compared to all arms in $\mathcal{A}$ for $q_r$ times. This step crucially protects the algorithm against cases where a sub-optimal arm becomes the candidate (and continues to become the candidate). For example, suppose $K = [5]$ and the arms are linearly ordered as $1 \succ 2 \succ \cdots \succ 5$. Furthermore suppose that in some round $r$, we have that (a) 2 defeats $3, 4, 5$ and (b) 1 (best arm) defeats 2 but not the others. So, 2 is the candidate in round $r$; if 1 is not compared to $3, 4, 5$, then 2 would continue to be the candidate (leading to high regret).

4. If, for any arm $j$, there is arm $i$ such that $\widehat{p}_{i,j}(r) > \frac{1}{2} + \gamma_{i,j}(r)$, then $j$ is eliminated from $\mathcal{A}$.

This continues until $T$ total comparisons are performed. See Algorithm 1 for a formal description. The main result of this section is to show that C2B achieves the guarantees stated in Theorems 1.1 and 1.2.

**Overview of the Analysis.** We provide a brief outline of the proofs of our main results. Towards proving Theorem 1.1, we first define two events:

- The first event, denoted $G$, ensures that $a^*$ is not eliminated during the execution of C2B. We show that $\mathbf{P}(G) \geq 1 - 1/T$.

- The second event, denoted $E(\delta)$, says that there exists a round $C(\delta)$ (defined later) such that for all $r > C(\delta)$, the estimate $\widehat{p}_{i,j}(r-1)$ satisfies the confidence interval of $c_{i,j}(r-1)$. Moreover, $\mathbf{P}(E(\delta)) \geq 1 - \delta$.

By union bound, $\mathbf{P}(G \cap E(\delta)) \geq 1 - \delta - 1/T$. Together, we use $G$ and $E(\delta)$ to argue that:

---

**Algorithm 1** C2B (CATCHING THE CONDORCET WINNER IN BATCHES)

1: **Input:** Arms $\mathcal{B}$, time-horizon $T$, integer $B \geq 1$
2: active arms $\mathcal{A} \leftarrow \mathcal{B}$, $r \leftarrow 1$, emprical probabilities $\widehat{p}_{i,j}(0) = \frac{1}{2}$ for all $i, j \in \mathcal{B}^2$
3: **while** number of comparisons $\leq T$ **do**
4:     **if** $\mathcal{A} = \{i\}$ for some $i$ **then** play $(i, i)$ for remaining trials
5:     $D_r(i) \leftarrow \{j \in \mathcal{A} : \widehat{p}_{i,j}(r-1) > \frac{1}{2} + c_{i,j}(r-1)\}$
6:     $i_r \leftarrow \arg\max_{i \in \mathcal{A}} |D_r(i)|$
7:     **for** $i \in \mathcal{A} \setminus \{i_r\}$ **do**
8:         **if** $i \in D_r(i_r)$ **then**
9:             compare $(i_r, i)$ for $q_r$ times
10:         **else**
11:             for each $j \in \mathcal{A}$, compare $(i, j)$ for $q_r$ times
12:     compute $\widehat{p}_{i,j}(r)$ values
13:     **if** $\exists i, j : \widehat{p}_{i,j}(r) > \frac{1}{2} + \gamma_{i,j}(r)$ **then**
14:         $\mathcal{A} \leftarrow \mathcal{A} \setminus \{j\}$
15:     $r \leftarrow r + 1$

---

- the best arm, $a^*$, is *not defeated* by any arm $i$ in any round $r > C(\delta)$ (formalized in Lemma 3.5),

- and that there exists a round $r(\delta) \geq C(\delta)$ such that for every round after $r(\delta)$, arm $a^*$ defeats *every other arm* (formalized in Lemma 3.7).

Intuitively, these observations imply that our algorithm identifies the best arm after $r(\delta)$ rounds. Thus, beyond round $r(\delta)$, we only perform pairwise comparisons of the form $(a^*, i)$ for $i \neq a^*$: thus, $a^*$ is used as an anchor to eliminate sub-optimal arms. We then analyze the regret in two parts: (i) regret incurred up to round $r(\delta)$, which is upper bounded by $K^2 \sum_{r \leq r(\delta)} q^r$ and (ii) regret after $r(\delta)$, which is the regret incurred in eliminating sub-optimal arms using $a^*$ as an anchor. Finally, we can use the high-probability bound to also obtain a bound on the expected regret, proving Theorem 1.2. We provide some details of the proof of Theorem 1.1. We defer the proof of Theorem 1.2 to Appendix D.

## 3.1 The Analysis

In this section, we give high-probability and expected regret bounds for C2B. Recall that $q = T^{1/B}$, and that $q \geq 2$. The following lemma is used to prove that $a^*$ is never eliminated. We defer the proofs of the Lemmas 3.1, 3.2 to Appendix B.

**Lemma 3.1.** *For any batch $r \in [B]$, and for any pair $(i, j)$, we have*

$$\mathbf{P}\left(|\widehat{p}_{i,j}(r) - p_{i,j}| > \gamma_{i,j}(r)\right) \leq 2\eta,$$

*where $\eta = 1/K^2 BT$.*

We first define the *good* event $G$ as follows.

**Definition 3.1** (Event $G$). *An estimate $\widehat{p}_{i,j}(r)$ at the end of batch $r$ is **strongly-correct** if $|\widehat{p}_{i,j}(r) - p_{i,j}| \leq \gamma_{i,j}(r)$. We say that event $G$ occurs if every estimate in every batch $r \in [B]$ is strongly-correct.*

The following two lemmas show that $G$ occurs with high probability and that $a^*$ is not eliminated under $G$.

**Lemma 3.2.** *The probability that every estimate in every batch of C2B is strongly-correct is at least $1 - 1/T$.*

**Lemma 3.3.** *Conditioned on $G$, $a^*$ is never eliminated from $\mathcal{A}$ in the elimination step of C2B.*

*Proof.* In C2B, an arm $j$ is deleted in batch $r$ iff there is an arm $i \in \mathcal{A}$ with $\widehat{p}_{i,j}(r) > \frac{1}{2} + \gamma_{i,j}(r)$. If $a^*$ is eliminated due to some arm $j$, then by definition of event $G$, we get $p_{j,a^*} \geq \widehat{p}_{i,j}(r) - \gamma_{i,j}(r) > \frac{1}{2}$, a contradiction. $\square$

### 3.1.1 High-probability Regret Bound

In this section, we give details required to prove Theorem 1.1. Fix any $\delta > 0$. We first define another good event as follows.

**Definition 3.2** (Event $E(\delta)$). *An estimate $\widehat{p}_{i,j}(r)$ in batch $r$ is **weakly-correct** if $|\widehat{p}_{i,j}(r) - p_{i,j}| \leq c_{i,j}(r)$. Let $C(\delta) := \lceil \frac{1}{2} \log_q(1/\delta) \rceil$. We say that event $E(\delta)$ occurs if for each batch $r \geq C(\delta)$, every estimate is weakly-correct.*

The next lemma shows that $E(\delta)$ occurs with probability at least $1 - \delta$.

**Lemma 3.4.** *For all $\delta > 0$, we have*

$$\mathbf{P}(\neg E(\delta)) \;=\; \mathbf{P}\left(\exists r \geq C(\delta), i, j : |\widehat{p}_{i,j}(r) - p_{i,j}| > c_{i,j}(r)\right) \;\leq\; \delta.$$

We will analyze our algorithm under both events $G$ and $E(\delta)$. Note that event $G$ is required to ensure that $a^*$ is not eliminated in rounds before $C(\delta)$ (where the Lemma 3.4 does not apply).

**Lemma 3.5.** *Conditioned on $G$ and $E(\delta)$, for any round $r > C(\delta)$, arm $a^*$ is not defeated by any other arm, i.e., $a^* \notin \cup_{i \neq a^*} D_r(i)$.*

To proceed, we need the following definitions.

**Definition 3.3.** *The candidate $i_r$ of round $r$ is called the **champion** if $|D_r(i_r)| = |\mathcal{A}| - 1$; that is, if $i_r$ defeats every other active arm.*

**Definition 3.4.** *Let $r(\delta) \geq C(\delta) + 1$ be the smallest integer such that*

$$q^{r(\delta)} \geq 2A \log A, \qquad where \; A := \frac{32}{\Delta_{\min}^2} \cdot \log(2K^2).$$

We use the following inequality based on this choice of $r(\delta)$.

**Lemma 3.6.** *The above choice of $r(\delta)$ satisfies*

$$q^r > \frac{8}{\Delta_{\min}^2} \cdot \log\left(2K^2 q_r\right), \qquad \forall r \geq r(\delta).$$

*Proof of Lemma 3.6.* Using the fact that $q_r \leq q^r$, it suffices to show $q^r \geq \frac{8}{\Delta_{\min}^2} \cdot \left(\log(2K^2) + \log q^r\right)$. Moreover,

$$\log(2K^2) + \log q^r \leq \left(1 + \log(2K^2)\right) \cdot (1 + \log q^r) \leq 4 \cdot \log(2K^2) \cdot \log q^r,$$

where the last inequality uses $K \geq 2$, $r \geq r(\delta) \geq 1$ and $q \geq 2$. So, it suffices to show:

$$q^r > A \cdot \log(q^r), \quad \forall r \geq r(\delta), \qquad where \; A = \frac{32}{\Delta_{\min}^2} \cdot \log(2K^2) \tag{3}$$

Below, let $x = q^r$, $R := 2A \log A$ and function $f(x) := x - A \log x$. We will show that $f(x) > 0$ for all $x \geq R$, which would imply (3) because $q^{r(\delta)} \geq R$. As $R \geq A$, and $f$ is increasing for $x \geq A$, it suffices to show that $f(R) \geq 0$. Indeed,

$$\frac{f(R)}{A} = 2 \log A - \log(2A \log A) = \log A - \log(2 \log A) > 0,$$

where the inequality uses $A \geq 8$. $\qquad \square$

Then, we have the following.

**Lemma 3.7.** *Conditioned on $G$ and $E(\delta)$, the best arm $a^*$ is the champion in every round $r > r(\delta)$.*

We now have all components required to prove Theorem 1.1; its proof, and the proofs of the aforementioned lemmas can be found in Appendix C.

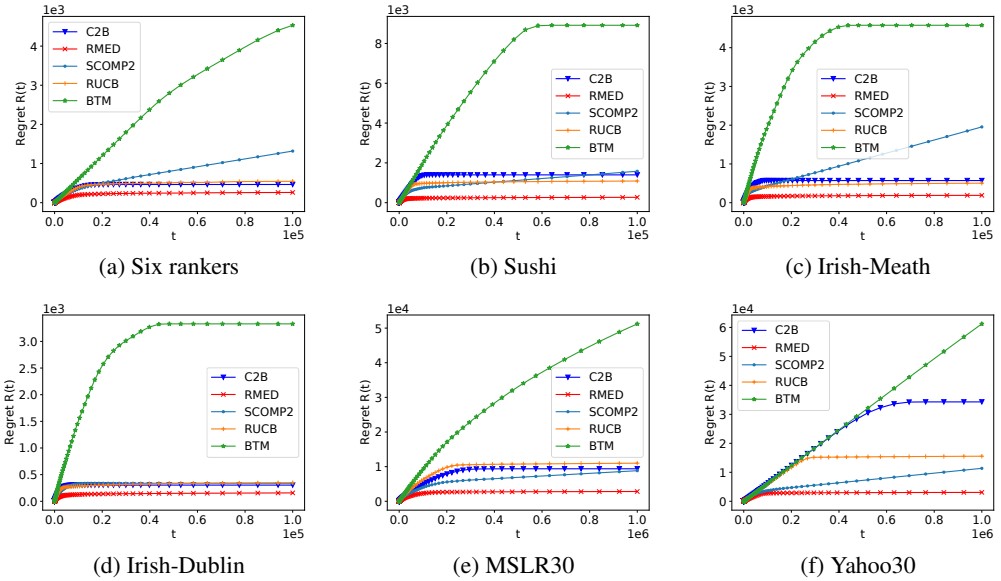

Figure 1: Regret v/s t plots of algorithms when $B = \lfloor \log(T) \rfloor$

# 4 Computational Results

In this section, we provide details of our computational experiments. The goal of our experiments is to answer the following questions: (i) How does the regret of C2B using $B = \lfloor \log(T) \rfloor$ batches compare to that of existing fully sequential as well as batched algorithms? and (ii) Can the regret of C2B match the regret of the best known sequential algorithms; if yes, then how many rounds suffice to achieve this? Towards answering (i), we compare C2B to a representative set of sequential algorithms for dueling bandits using the library due to [35]. We compare C2B to the sequential algorithms RUCB [54], RMED [35], and BEAT-THE-MEAN (BTM) [51]. We allow these algorithms to work as prescribed; that is, they work in $B = T$ batches. The reason that we chose these sequential algorithms is that our batched algorithm (C2B) is based on a similar paradigm, and such a comparison demonstrates the power of adaptivity in this context. We also compare C2B to the batched algorithm SCOMP2 [2]. We plot the cumulative regret $R(t)$ incurred by the algorithms against time $t$. We set $B = \lfloor \log(T) \rfloor$ for C2B and SCOMP2 in this experiment. For (ii), we increased $B$ by a small amount; we found that the performance of C2B improves noticeably when given a constant number of additional rounds (we use $B = \lfloor \log(T) \rfloor + 6$ in this experiment). We perform these experiments using the following real-world datasets.

**Six rankers.** This dataset is based on the 6 retrieval functions used in the engine of ArXiv.org.

**Sushi.** The Sushi dataset is based on the Sushi preference dataset [34] that contains the preference data regarding 100 types of Sushi. A preference dataset using the top-16 most popular types of sushi is obtained.

**Irish election data.** The Irish election data for Dublin and Meath is available at *preflib.org*. It contains partial preference orders over candidates. As in [3], these are transformed into preference matrices by selecting a subset of candiates to ensure that a Condorcet winner exists. There are 12 candidates in the **Irish-Meath** dataset, and 8 in the **Irish-Dublin** dataset.

**MSLR and Yahoo! data.** We also run experiments on two web search ranking datasets: the Microsoft Learning to Rank (MSLR) dataset [40] and the Yahoo! Learning to Rank Challenge Set 1 [16]. These datasets have been used in prior work on online ranker evaluation [53, 37]. We use preference matrices generated using the "navigational" configuration (see [37] for details). The MSLR dataset has 136 rankers and the Yahoo! dataset has 700 rankers. We sample 30 rankers from each dataset while ensuring the existence of a Condorcet winner. In this way, we obtain two datasets, denoted **MSLR30** and **Yahoo30**.

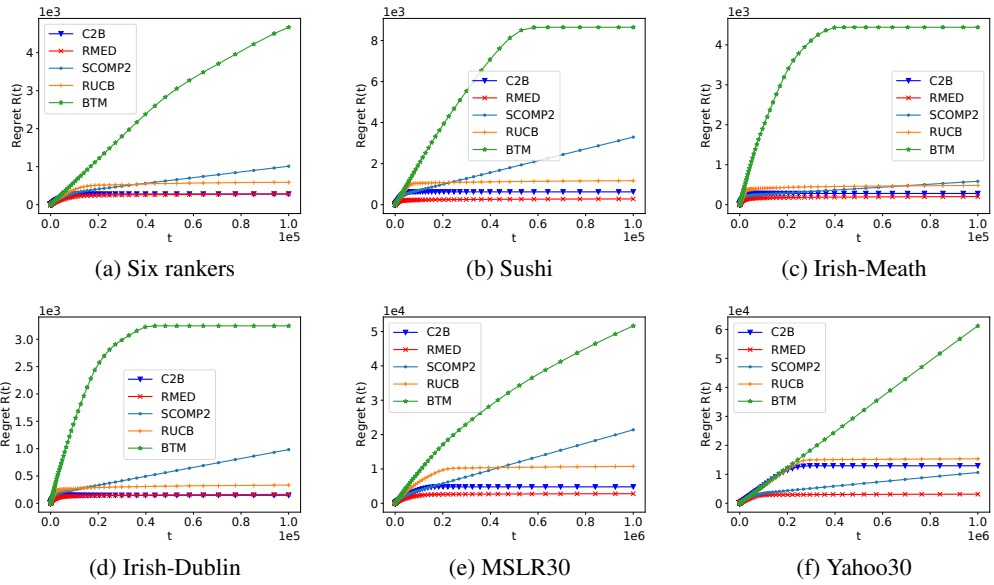

Figure 2: Regret v/s t plots of algorithms when $B = \lfloor \log(T) \rfloor + 6$

Note that there exists a Condorcet winner in all datasets. We repeat each experiment 20 times and report the average regret. In our algorithm, we use the *KL-divergence based confidence bound* due to [35] for elimination as it performs much better empirically, and our theoretical bounds continue to hold (see §E). This KL-divergence based elimination criterion eliminates an arm $i$ in round $r$ if $I_i(r) - I^*(r) > \log(T) + f(K)$ where $I_i(r) = \sum_{j:\widehat{p}_{i,j}(r) < \frac{1}{2}} N_{i,j}(r) \cdot D_{\mathrm{KL}}(\widehat{p}_{i,j}(r), \frac{1}{2})$ and $I^*(r) = \min_{j \in [K]} I_i(r)$.

**Computational Results.** As mentioned earlier, we compare our algorithms against a representative set of sequential dueling bandits algorithms (RUCB, RMED, and BTM). We set $\alpha = 0.51$ for RUCB, and $f(K) = 0.3K^{1.01}$ for RMED and C2B, and $\gamma = 1.3$ for BTM: these parameters are known to perform well both theoretically and empirically [35]. We set $T = 10^6$ for MSLR30 and Yahoo30 datasets (as they have larger number of arms), and $T = 10^5$ for the remaining four. For the first set of experiments, we set $B = \lfloor \log(T) \rfloor$. We observe that C2B always outperforms BTM and beats SCOMP2 on most of the datasets. We observe that even when SCOMP2 beats C2B it has a slightly linear curve (implying that its regret would keep increasing as $T$ increases) while the regret curve of C2B is mostly flat. Furthermore, C2B performs comparably to RUCB in all datasets except Yahoo30. We plot the results in Figure 1. In the second set of experiments, we set $B = \lfloor \log(T) \rfloor + 6$. We observe that C2B always outperforms RUCB and, in fact, performs comparably to RMED on all datasets except Yahoo30. We plot the results in Figure 2. Finally, we note that SCOMP2 exhibits varying performance across runs (even on the same dataset) and we think that this is due to the randomness involved in selecting the "seed set".

## 5 Conclusion

In this paper, we proposed a batched algorithm, named C2B, for the $K$-armed dueling bandit problem. Assuming the existence of a Condorcet winner, we show both high-probability and expected regret bounds for C2B that trade-off smoothly with the number of batches. Furthermore, we obtain asymptotic regret of $O(K^2 \log^2(K)) + O(K \log(T))$ in $O(\log(T))$ batches, nearly matching the best regret bounds known in the fully sequential setting under the Condorcet condition. Our computational results show that C2B, using $O(\log(T))$ batches, achieves almost the same performance as fully sequential algorithms over a variety of real-world datasets. A direction for future research is to design batched algorithms for the $K$-armed dueling bandit problem when a Condorcet winner does not exist; for example, designing an algorithm for a more general concept of winner, such as *Copeland winner* [48] or *von Neumann winner* [22].

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
