# A  Further Related Work

**Batched algorithms for multi-armed bandits.**    Batched processing for the stochastic multi-armed bandit problem has been investigated in the past few years. A special case when there are two bandits was studied by [39]. They obtain a worst-case regret bound of $O\left(\left(\frac{T}{\log(T)}\right)^{1/B}\frac{\log(T)}{\Delta_{\min}}\right)$. [25] studied the general problem and obtained a worst-case regret bound of $O\left(\frac{K\log(K)T^{1/B}\log(T)}{\Delta_{\min}}\right)$, which was later improved by [23] to $O\left(\frac{KT^{1/B}\log(T)}{\Delta_{\min}}\right)$. Furthermore, [23] obtained an instance-dependent regret bound of $\sum_{j:\Delta_j>0}T^{1/B}O\left(\frac{\log(T)}{\Delta_j}\right)$. Our results for batched dueling bandits are of a similar flavor; that is, we get a similar dependence on $T$ and $B$. [23] also give batched algorithms for stochastic linear bandits and adversarial multi-armed bandits.

**Adaptivity in combinatorial optimization.**    Adaptivity and batch processing has been recently studied for stochastic submodular cover [27, 1, 24, 26], and for various stochastic "maximization" problems such as knapsack [20, 13], matching [10, 12], probing [30] and orienteering [28, 29, 11]. Recently, there have also been several results examining the role of adaptivity in (deterministic) submodular optimization; e.g. [8, 6, 9, 7, 17].

# B  Missing Proofs from §3.1

*Proof of Lemma 3.1.*  Note that $\mathbb{E}[\widehat{p}_{i,j}(r)] = p_{i,j}$, and applying Hoeffding's inequality gives

$$\mathbf{P}\left(|\widehat{p}_{i,j}^{\,r+1} - p_{i,j}| > \gamma_{i,j}(r)\right) \le 2\exp\left(-2N_{i,j}(r)\cdot\gamma_{i,j}(r)^2\right) \le 2\eta.$$

$\square$

*Proof of Lemma 3.2.*  Applying Lemma 3.1 and taking a union bound over all pairs and batches, we get that the probability that some estimate is incorrect is at most $\binom{K}{2}\times B\times 2\eta \le \frac{1}{T}$ where $\eta = 1/K^2BT$. Thus, $\mathbf{P}(\overline{G}) \le \frac{1}{T}$. $\square$

*Proof of Lemma 3.3.*  In C2B, an arm $j$ is deleted in batch $r$ iff there is an arm $i \in \mathcal{A}$ with $\widehat{p}_{i,j}(r) > \frac{1}{2} + 2\gamma_{i,j}(r)$. If $a^*$ is eliminated due to some arm $j$, then by definition of event $G$, we get $p_{j,a^*} > \frac{1}{2} + \gamma_{i,j}(r) > \frac{1}{2}$, a contradiction. $\square$

# C  Missing Proofs from §3.1.1

*Proof of Lemma 3.4.*  For any pair $i, j$ of arms and round $r$, let $B_{i,j}(r)$ denote the event that $|\widehat{p}_{i,j}(r) - p_{i,j}| > c_{i,j}(r)$. Note that $N_{ij}(r) \le \sum_{s=1}^{r}q_s \le 2q_r$. For any integer $n$, let $s_{ij}(n)$ denote the sample average of $n$ independent Bernoulli r.v.s with probability $p_{ij}$. By Hoeffding's bound,

$$\mathbf{P}[|s_{ij}(n) - p_{ij}| > c] \le 2e^{-2nc^2}, \qquad \text{for any } c \in [0,1].$$

We now bound

$$\mathbf{P}[B_{ij}(r)] \le \sum_{n=0}^{2q_r}\mathbf{P}[B_{ij}(r) \wedge N_{ij}(r) = n]$$

$$\le \sum_{n=0}^{2q_r}\mathbf{P}\left[|s_{ij}(n) - p_{ij}| > \sqrt{\frac{2\log(2K^2q_r)}{n}}\right] \le \sum_{n=0}^{2q_r}2\exp\left(-2n\cdot\frac{2\log(2K^2q_r)}{n}\right)$$

$$\le 4q_r\cdot\frac{1}{(2K^2q_r)^4} \le \frac{1}{4K^2\cdot q_r^2}$$

The second inequality uses the definition of $c_{ij}(r)$ when $N_{ij}(r) = n$. The last inequality uses $K \geq 2$. Next, by a union bound over arms and rounds, we can write the desired probability as

$$
\begin{aligned}
\mathbf{P}(\exists r \geq C(\delta), i, j : B_{i,j}(r)) &\leq \sum_{r \geq C(\delta)} \sum_{i < j} \mathbf{P}(B_{i,j}(r)) \\
&\leq \sum_{r \geq C(\delta)} \binom{K}{2} \cdot \frac{1}{4K^2 \cdot q_r^2} \leq \sum_{r \geq C(\delta)} \frac{1}{8q_r^2} \\
&\leq \sum_{r \geq C(\delta)} \frac{1}{2q^{2r}} = \frac{1}{2q^{2C(\delta)}} \cdot \left(1 + \frac{1}{q^2} + \frac{1}{q^4} + \cdots \right) \leq \frac{1}{q^{2C(\delta)}} \leq \delta
\end{aligned}
\tag{4}
$$

The second inequality above uses the bound on $\mathbf{P}[B_{ij}(r)]$. The first inequality in (4) uses $q_r = \lfloor q^r \rfloor \geq q^r - 1 \geq \frac{q^r}{2}$ as $q \geq 2$. The last inequality in (4) uses the definition of $C(\delta)$.

The lemma now follows by the definition of event $\neg E(\delta)$ as $\exists r \geq C(\delta), i, j : B_{i,j}(r)$. $\qquad \square$

*Proof of Lemma 3.5.* Fix any round $r \geq C(\delta) + 1$. Suppose that $a^* \in D_r(i)$ for some other arm $i$. This implies that $\widehat{p}_{i,a^*}(r-1) > \frac{1}{2} + c_{i,a^*}(r-1)$. But under event $E(\delta)$, we have $|\widehat{p}_{i,a^*}(r-1) - p_{i,a^*}| \leq c_{i,a^*}(r-1)$ because $r - 1 \geq C(\delta)$. Combined, these two observations imply $p_{i,a^*} > \frac{1}{2}$, a contradiction. $\qquad \square$

*Proof of Lemma 3.7.* We first argue that $a^*$ is compared to all active arms in each round $r \geq r(\delta)$. By Lemma 3.3, we know $a^* \in \mathcal{A}$. By Lemma 3.5, we have $a^* \notin D_r(j)$ for all $j \neq a^*$ because $r \geq r(\delta) \geq 1 + C(\delta)$. If candidate $i_r \neq a^*$, then $a^*$ will be compared to all $j \in \mathcal{A}$ (since $a^* \notin D_r(i_r)$). On the other hand, if $i_r = a^*$, then (1) for any $j \in D_r(a^*)$, arm $j$ is only compared to $a^*$, and (2) for any $j \in \mathcal{A} \setminus D_r(a^*)$, arm $j$ is compared to all active arms including $a^*$.

Next, we show that for any round $r \geq r(\delta) + 1$, arm $a^*$ defeats all other arms, i.e., $|D_r(a^*)| = |\mathcal{A}| - 1$. This would imply that $i_r = a^*$ and $a^*$ is the champion. Consider any arm $j \in \mathcal{A} \setminus a^*$. Since $a^*$ is compared to all active arms in round $r - 1 \geq r(\delta)$, we have

$$
N_{a^*,j}(r-1) \geq q^{r-1} > \frac{8}{\Delta_{\min}^2} \cdot \log\left(2K^2 q_{r-1}\right),
$$

where the second inequality is by Lemma 3.6 with $r - 1 \geq r(\delta)$. Now, by definition, we have

$$
c_{a^*,j}(r-1) = \sqrt{\frac{2\log\left(2K^2 q_{r-1}\right)}{N_{a^*,j}(r-1)}} < \sqrt{\frac{2\log\left(2K^2 q_{r-1}\right)}{\frac{8}{\Delta_{\min}^2}\log\left(2K^2 q_{r-1}\right)}} = \frac{\Delta_{\min}}{2}.
$$

Given this, it is easy to show that $a^*$ defeats arm $j$ in round $r$. Conditioned on $E(\delta)$, we know that $|\widehat{p}_{a^*,j}(r-1) - p_{a^*,j}| \leq c_{a^*,j}(r-1) \leq \frac{\Delta_{\min}}{2}$. Then, we have

$$
\widehat{p}_{a^*,j}(r-1) \geq p_{a^*,j} - \frac{\Delta_{\min}}{2} = \frac{1}{2} + \Delta_j - \frac{\Delta_{\min}}{2} \geq \frac{1}{2} + \frac{\Delta_{\min}}{2} > \frac{1}{2} + c_{a^*,j}(r-1).
$$

Therefore, $j \in D_r(a^*)$. It now follows that for any round $r \geq r(\delta) + 1$, arm $a^*$ is the champion. $\qquad \square$

*Proof of Theorem 1.1.* First, recall that in round $r$ of C2B, any pair is compared $q_r = \lfloor q^r \rfloor$ times where $q = T^{1/B}$. Since $q^B = T$, C2B uses at most $B$ rounds.

For the rest of proof, we fix $\delta > 0$. We now analyze the regret incurred by C2B, conditioned on events $G$ and $E(\delta)$. Recall that $\mathbf{P}(G) \geq 1 - 1/T$ (Lemma 3.2), and $\mathbf{P}(E(\delta)) \geq 1 - \delta$ (Lemma 3.4). Thus, $\mathbf{P}(G \cap E(\delta)) \geq 1 - \delta - 1/T$. Let $R_1$ and $R_2$ denote the regret incurred before and after round $r(\delta)$ (see Definition 3.4) respectively.

**Bounding $R_1$.** This is the regret incurred up to (and including) round $r(\delta)$. We upper bound the regret by considering all pairwise comparisons every round $r \leq r(\delta)$.

$$R_1 \;\leq\; K^2 \cdot \sum_{r \leq r(\delta)} q_r \;\leq\; K^2 \cdot \sum_{r \leq r(\delta)} q^r \;\leq\; 2K^2 \cdot q^{r(\delta)}$$

$$\leq 2K^2 \cdot \max\left\{ q \cdot 2A \log A \,,\, q^{C(\delta)+1} \right\},$$

where the last inequality uses Definition 3.4, recall $A = \frac{16}{\Delta_{\min}^2} \cdot \log(2K^2)$. Plugging in the value of $C(\delta) \leq 1 + \frac{1}{2} \log_q(1/\delta)$, we obtain

$$R_1 \leq O(K^2) \cdot \max\left\{ q \cdot \frac{\log K}{\Delta_{\min}^2} \cdot \log\left(\frac{\log K}{\Delta_{\min}}\right) \,,\, q^2 \sqrt{\frac{1}{\delta}} \right\}. \tag{5}$$

**Bounding $R_2$.** This is the regret in rounds $r \geq r(\delta) + 1$. By Lemma 3.7, arm $a^*$ is the champion in all these rounds. So, the only comparisons in these rounds are of the form $(a^*, j)$ for $j \in \mathcal{A}$.

Consider any arm $j \neq a^*$. Let $T_j$ be the total number of comparisons that $j$ participates in after round $r(\delta)$. Let $r$ be the penultimate round that $j$ is played in. We can assume that $r \geq r(\delta)$ (otherwise arm $j$ will never participate in rounds after $r(\delta)$, i.e., $T_j = 0$). As arm $j$ is *not* eliminated after round $r$,

$$\widehat{p}_{a^*,j}(r) \leq \frac{1}{2} + \gamma_{a^*,j}(r).$$

Moreover, by $E(\delta)$, we have $\widehat{p}_{a^*,j}(r) \geq p_{a^*,j} - c_{a^*,j}(r)$ because $r \geq r(\delta) \geq C(\delta)$. So,

$$\frac{1}{2} + \Delta_j = p_{a^*,j} \leq \widehat{p}_{a^*,j}(r) + c_{a^*,j}(r) \leq \frac{1}{2} + \gamma_{a^*,j}(r) + c_{a^*,j}(r).$$

It follows that

$$\Delta_j \leq \gamma_{a^*,j}(r) + c_{a^*,j}(r) \leq \frac{3}{\sqrt{2}} \sqrt{\frac{\log(2K^2 BT)}{N_{a^*,j}(r)}}$$

where the final inequality follows by definition of $c$ and $\gamma$. On re-arranging, we get $N_{a^*,j}(r) \leq \frac{9 \log(2K^2 BT)}{2\Delta_j^2}$. As $r + 1$ is the last round that $j$ is played in, and $j$ is only compared to $a^*$ in each round after $r(\delta)$,

$$T_j \;\leq\; N_{a^*,j}(r+1) \;\leq\; N_{a^*,j}(r) + 2q \cdot N_{a^*,j}(r) \;\leq\; \frac{15q \cdot \log(2K^2 BT)}{\Delta_j^2}.$$

The second inequality follows since $j$ is compared to $a^*$ in rounds $r$ and $r + 1$, and the number of comparisons in round $r + 1$ is $\lfloor q^{r+1} \rfloor \leq q \cdot (2q_r) \leq 2q \cdot N_{a^*,j}(r)$. Adding over all arms $j$, the total regret accumulated beyond round $r(\delta)$ is

$$R_2 = \sum_{j \neq a^*} T_j \Delta_j \leq \sum_{j \neq a^*} O\left(\frac{q \cdot \log(KT)}{\Delta_j}\right). \tag{6}$$

Combining (5) and (6), and using $q = T^{1/B}$, we obtain

$$R(T) \leq O\left(T^{1/B} \cdot \frac{K^2 \log(K)}{\Delta_{\min}^2} \cdot \log\left(\frac{\log K}{\Delta_{\min}}\right)\right) + O\left(T^{2/B} \cdot K^2 \cdot \sqrt{\frac{1}{\delta}}\right)$$

$$+ \sum_{j \neq a^*} O\left(\frac{T^{1/B} \cdot \log(KT)}{\Delta_j}\right).$$

This completes the proof Theorem 1.1. $\qquad\square$

# D  Expected Regret Bound

In this section, we present the proof of Theorem 1.2. We first state the definitions needed in the proof. Let $F_X(\cdot)$ denote the cumulative density function (CDF) of a random variable $X$; that is, $F_X(x) = \mathbf{P}(X \leq x)$. The inverse CDF of $X$, denoted $F_X^{-1}$, is defined as $F_X^{-1}(z) = \inf\{x : \mathbf{P}(X \leq x) \geq z\}$ where $z \in [0, 1]$. We will use the identity $\mathbb{E}[X] = \int_0^1 F_X^{-1}(z)dz$.

*Proof of Theorem 1.2.* First, note that in round $r$ of C2B, any pair is compared $q_r = \lfloor q^r \rfloor$ times where $q = T^{1/B}$. Since $q^B = T$, C2B uses at most $B$ rounds.

Let $R(T)$ be the random variable denoting the regret incurred by C2B. By Theorem 1.1, we know that with probability at least $1 - \delta - 1/T$,

$$R(T) \leq O\left(T^{1/B} \cdot \frac{K^2 \log(K)}{\Delta_{\min}^2} \cdot \log\left(\frac{\log K}{\Delta_{\min}}\right)\right) + O\left(T^{2/B} \cdot K^2 \cdot \sqrt{\frac{1}{\delta}}\right)$$
$$+ \sum_{j \neq a^*} O\left(\frac{T^{1/B} \cdot \log(KT)}{\Delta_j}\right).$$

Thus, $F_{R(T)}^{-1}(1 - \delta - 1/T) \leq G(\delta)$ where

$$G(\delta) := A + O\left(T^{2/B} \cdot K^2 \cdot \sqrt{\frac{1}{\delta}}\right) + B$$

where to simplify notation we set $A = O\left(T^{1/B} \cdot \frac{K^2 \log(K)}{\Delta_{\min}^2} \cdot \log\left(\frac{\log K}{\Delta_{\min}}\right)\right)$ and $B = \sum_{j \neq a^*} O\left(\frac{T^{1/B} \cdot \log(KT)}{\Delta_j}\right)$. Using the identity for expectation of a random variable, we get

$$\mathbb{E}[R(T)] = \int_0^1 F_{R(T)}^{-1}(z)dz$$
$$= \int_0^{1-\frac{1}{T}} F_{R(T)}^{-1}(z)dz + \underbrace{\int_{1-\frac{1}{T}}^T F_{R(T)}^{-1}(z)dz}_{\leq T \cdot \frac{1}{T} = 1}$$
$$\leq \int_0^{1-\frac{1}{T}} F_{R(T)}^{-1}(z)dz \; + \; 1$$
$$= \int_{1-\frac{1}{T}}^0 F_{R(T)}^{-1}\left(1 - \delta - \frac{1}{T}\right)(-d\delta) \; + \; 1$$
$$\leq \int_0^{1-\frac{1}{T}} G(\delta)d\delta \; + \; 1$$
$$\leq A + O\left(T^{2/B} \cdot K^2\right) + B + 1$$

where the fourth equality follows by setting $1 - q - 1/T = \delta$ and the final inequality follows since $\int_0^1 \left(\frac{1}{\delta}\right)^{1/2} \leq 2$. Thus,

$$\mathbb{E}[R(T)] \leq O\left(T^{1/B} \cdot \frac{K^2 \log(K)}{\Delta_{\min}^2} \cdot \log\left(\frac{\log K}{\Delta_{\min}}\right)\right) + O\left(T^{2/B} \cdot K^2\right)$$
$$+ \sum_{j \neq a^*} O\left(\frac{T^{1/B} \cdot \log(KT)}{\Delta_j}\right).$$

This completes the proof of Theorem 1.2. □

# E  The Batched Algorithm with KL-based Elimination Criterion

In this section, we modify C2B to use a Kullback-Leibler divergence based elimination criterion. We provide a complete description of the algorithm, denoted C2B-KL, in Algorithm 2. In what follows, we highlight the main differences of C2B-KL from C2B. Recall the following notation. We use $\mathcal{A}$ to denote the current set of *active* arms; i.e., the arms that have not been eliminated. We use index $r$ for rounds or batches. If pair $(i, j)$ is compared in round $r$, it is compared $q_r = \lfloor q^r \rfloor$ times where $q = T^{1/B}$. We define the following quantities at the *end* of each round $r$:

- $N_{i,j}(r)$ is the total number of times the pair $(i, j)$ has been compared.
- $\widehat{p}_{i,j}(r)$ is the frequentist estimate of $p_{i,j}$, i.e.,

$$\widehat{p}_{i,j}(r) = \frac{\# \ i \ \text{wins against} \ j \ \text{until end of round} \ r}{N_{i,j}(r)} . \tag{7}$$

- A confidence-interval radius for each $(i, j)$ pair:

$$c_{i,j}(r) = \sqrt{\frac{2 \log(2K^2 q_r)}{N_{i,j}(r)}}$$

- We define a term $I_j(r)$ which, at a high-level, measures how unlikely it is for $j$ to be the Condorcet winner at the end of batch $r$:

$$I_j(r) = \sum_{i:\widehat{p}_{i,j}(r) \geq \frac{1}{2}} D_{\mathrm{KL}}\left(\widehat{p}_{i,j}(r), \frac{1}{2}\right) \cdot N_{i,j}(r),$$

where $D_{\mathrm{KL}}(p, q)$ denotes the Kullback–Leibler divergence between two Bernoulli distributions: $B(p)$ and $B(q)$. We define $I^*(r) = \min_{j \in \mathcal{A}} I_j(r)$.

The $B$-round algorithm, C2B-KL, proceeds exactly as C2B. The only change is in the *elimination criterion*, which we describe next.

**Elimination Criterion.**  In round $r$, if, for any arm $j$, we have $I_j(r) - I^*(r) > \log(T) + f(K)$, then $j$ is eliminated from $\mathcal{A}$. Here $f(K)$ is a non-negative function of $K$, independent of $r$.

The main result of this section is to show that C2B-KL achieves the following guarantee.

**Theorem E.1.** *For any integer $B \geq 1$, there is an algorithm for the $K$-armed dueling bandit problem that uses at most $B$ rounds. Furthermore, for any $\delta > 0$, with probability at least $1 - \delta - \frac{1}{T} \cdot e^{K \log(C) - f(K)}$, where $C$ is some constant (see Lemma E.2), its regret under the Condorcet condition is at most*

$$R(T) \leq O\left(T^{1/B} \cdot \frac{K^2 \log(K)}{\Delta_{\min}^2} \cdot \log\left(\frac{\log K}{\Delta_{\min}}\right)\right) \quad + O\left(T^{2/B} \cdot K^2 \cdot \sqrt{\frac{1}{\delta}}\right) + \sum_{j \neq a^*} O\left(\frac{T^{1/B} \cdot \log(T)}{\Delta_j}\right)$$

$$+ \sum_{j \neq a^*} O\left(\frac{T^{1/B} \cdot f(K)}{\Delta_j}\right)$$

**Remark.**  Setting $f(K) > K \log(C)$, we get the same asymptotic expected regret bound as in Theorem 1.2. Following [35], we set $f(K) = 0.3K^{1.01}$ in our experiments.

We require the following result in the proof of Theorem E.1.

**Fact E.1.** *For any $\mu$ and $\mu_2$ satisfying $0 < \mu_2 < \mu < 1$. Let $C_1(\mu, \mu_2) = (\mu - \mu_2)^2/(2\mu(1 - \mu_2))$. Then, for any $\mu_3 \leq \mu_2$,*

$$D_{KL}(\mu_3, \mu) - D_{KL}(\mu_3, \mu_2) \geq C_1(\mu, \mu_2) > 0.$$

The high-level outline of the analysis is exactly the same as that of C2B. For completeness, we provide the analysis in the following section; however, we skip the proofs of lemmas that follow from the analysis of C2B.

---
**Algorithm 2** C2B-KL
---

1: **Input:** Arms $\mathcal{B}$, time-horizon $T$, integer $B \geq 1$
2: active arms $\mathcal{A} \leftarrow \mathcal{B}$, $r \leftarrow 1$, emprical probabilities $\widehat{p}_{i,j}(0) = \frac{1}{2}$ for all $i,j \in \mathcal{B}^2$
3: **while** number of comparisons $\leq T$ **do**
4:      **if** $\mathcal{A} = \{i\}$ for some $i$ **then** play $(i,i)$ for remaining trials
5:      $D_r(i) \leftarrow \{j \in \mathcal{A} : \widehat{p}_{i,j}(r-1) > \frac{1}{2} + c_{i,j}(r-1)\}$
6:      $i_r \leftarrow \arg\max_{i \in \mathcal{A}} |D_r(i)|$
7:      **for** $i \in \mathcal{A} \setminus \{i_r\}$ **do**
8:          **if** $i \in D_r(i_r)$ **then**
9:              compare $(i_r, i)$ for $q_r$ times
10:         **else**
11:             for each $j \in \mathcal{A}$, compare $(i,j)$ for $q_r$ times
12:      compute $\widehat{p}_{i,j}(r)$ values
13:      **if** $\exists j : I_j(r) - I^*(r) > \log(T) + f(K)$ **then**
14:         $\mathcal{A} \leftarrow \mathcal{A} \setminus \{j\}$
15:      $r \leftarrow r + 1$

---

## E.1 The Analysis

In this section, we prove the high-probability regret bound for C2B-KL. Recall that $q = T^{1/B}$, and that $q \geq 2$. We first show that, with high probability, $a^*$ is not eliminated during the execution of the algorithm. The following lemma formalizes this.

**Lemma E.2.** *Let $G$ denote the event that the best arm $a^*$ is not eliminated during the execution of* C2B-KL. *We can bound the probability of $\overline{G}$ as follows.*

$$\mathbf{P}(\overline{G}) \leq \frac{1}{T} \cdot e^{K \log(C) - f(K)},$$

*where $C = \max_j C(j) + 1$, is a constant, with $C(j) = \left( \dfrac{1}{e^{D_{KL}\left(p_{j,a^*},1/2\right)} - 1} + \dfrac{e^{C_1\left(p_{a^*,j},1/2\right)}}{\left(e^{C_1\left(p_{a^*,j},1/2\right)} - 1\right)^2} \right).$*

*Proof.* Let $n_j$ denote the number of times $a^*$ and $j$ are compared. Let $\widehat{p}_{a^*,j}(n_j)$ denote the frequentist estimate of $p_{a^*,j}$ when $a^*$ and $j$ are compared $n_j$ times (we will abuse notation and use $\widehat{p}_{a^*,j}$ when $n_j$ is clear from context). Let $S \in 2^{[K] \setminus \{a^*\}} \setminus \emptyset$, and consider vector $\{n_j \in \mathbb{N} : j \in S\}$. We define $A = \sum_{j \in S} D_{\text{KL}}(\widehat{p}_{j,a^*}, 1/2) \cdot n_j$. Let $D(S; \{n_j : j \in S\})$ denote the event that $a^*$ and $j$ are compared $n_j$ times and $\widehat{p}_{a^*,j} \leq 1/2$ for all $j \in S$, and that $A > \log(T) + f(K)$. The probability of this event upper bounds the probability that $a^*$ is eliminated (as per our elimination criterion) when $a^*$ and $j$ are compared $n_j$ times, and $\widehat{p}_{a^*,j} \leq 1/2$ for all $j \in S$. We will show that

$$\mathbf{P}(D(S; \{n_j : j \in S\})) \leq \frac{e^{-f(K)}}{T} \prod_{j \in S} \left( e^{-n_j D_{\text{KL}}\left(p_{j,a^*}, 1/2\right)} + n_j e^{C_1\left(p_{j,a^*}, 1/2\right)} \right) \qquad (8)$$

where $C_1(\mu_1, \mu_2) = (\mu_1 - \mu_2)^2/(2\mu_1(1 - \mu_2))$. Using the above, we first show that by taking a union bound over all $S \in 2^{[K]\setminus\{a^*\}} \setminus \emptyset$ and $\{n_j : j \in S\}$, we obtain the final result. We have

$$
\begin{aligned}
\mathbf{P}(\overline{G}) &\leq \sum_{S \in 2^{[K]\setminus\{a^*\}}\setminus\emptyset} \sum_{n_j \in \mathbb{N}^{|S|}} \mathbf{P}(D(S; \{n_j : j \in S\})) \\
&\leq \sum_{S \in 2^{[K]\setminus\{a^*\}}\setminus\emptyset} \sum_{n_j \in \mathbb{N}^{|S|}} \frac{e^{-f(K)}}{T} \prod_{j \in S} \left( e^{-n_j D_{\mathrm{KL}}\left(p_{j,a^*}, 1/2\right)} + n_j e^{C_1\left(p_{j,a^*}, 1/2\right)} \right) \\
&= \frac{e^{-f(K)}}{T} \sum_{S \in 2^{[K]\setminus\{a^*\}}\setminus\emptyset} \prod_{j \in S} \sum_{n_j \in \mathbb{N}} \left( e^{-n_j D_{\mathrm{KL}}\left(p_{j,a^*}, 1/2\right)} + n_j e^{C_1\left(p_{j,a^*}, 1/2\right)} \right) \qquad (9) \\
&= \frac{e^{-f(K)}}{T} \sum_{S \in 2^{[K]\setminus\{a^*\}}\setminus\emptyset} \prod_{j \in S} \left( \frac{1}{e^{D_{\mathrm{KL}}\left(p_{j,a^*}, 1/2\right)} - 1} + \frac{e^{C_1\left(p_{j,a^*}, 1/2\right)}}{\left(e^{C_1\left(p_{j,a^*}, 1/2\right)} - 1\right)^2} \right) \qquad (10) \\
&\leq \frac{e^{-f(K)}}{T} \sum_{S \in 2^{[K]\setminus\{a^*\}}\setminus\emptyset} (C - 1)^{|S|} \leq \frac{e^{-f(K)}}{T} \cdot C^K \qquad (11) \\
&= \frac{1}{T} \cdot e^{K \log(C) - f(K)}
\end{aligned}
$$

where (9) follows by swapping the order of summation and multiplication, (10) uses $\sum_{n=1}^{\infty} e^{-nx} = 1/(e^x - 1)$ and $\sum_{n=1}^{\infty} n e^{-nx} = e^x/(e^x - 1)^2$, and (11) follows by letting

$$
C(j) = \left( \frac{1}{e^{D_{\mathrm{KL}}\left(p_{j,a^*}, 1/2\right)} - 1} + \frac{e^{C_1\left(p_{j,a^*}, 1/2\right)}}{\left(e^{C_1\left(p_{j,a^*}, 1/2\right)} - 1\right)^2} \right), C = \max_j C(j) + 1 \text{ and the binomial theorem.}
$$

To complete the proof, we need to prove (8).

For the remainder of this proof, we fix $S \in 2^{[K]\setminus\{a^*\}} \setminus \emptyset$, and vector $\{n_j \in \mathbb{N} : j \in S\}$. Observe that

$$
\mathbf{P}(D(S; \{n_j : j \in S\})) = \mathbf{P}\left(A > \log(T) + f(K)\right) = \mathbf{P}\left(T < e^{-f(K)} \cdot e^A\right)
$$

where we defined $A = \sum_{j \in S} D_{\mathrm{KL}}\left(\widehat{p}_{j,a^*}, 1/2\right) \cdot n_j$. By Markov's inequality, we have

$$
\mathbf{P}\left(e^{-f(K)} \cdot e^A > T\right) \leq \frac{\mathbb{E}[e^{-f(K)} \cdot e^A]}{T} = \frac{e^{-f(K)}}{T} \cdot \mathbb{E}[e^A] \qquad (12)
$$

where the last equality follows since $f(K)$ is constant (with respect to $\{n_j\}$ values). So, it suffices to bound $\mathbb{E}[e^A]$. Towards this end, we define the following term:

$$
P_j(x_j) = \mathbf{P}\left(\widehat{p}_{j,a^*} \geq \frac{1}{2} \text{ and } D_{\mathrm{KL}}\left(\widehat{p}_{j,a^*}, \frac{1}{2}\right) \geq x_j\right).
$$

Then, we have

$$\mathbb{E}[e^A] = \int_{\{x_j\}\in[0,\log(2)]^{|S|}} \exp\left(\sum_{j\in S} n_j x_j\right) \prod_{j\in S} d(-P_j(x_j))$$

$$= \prod_{j\in S} \int_{x_j\in[0,\log 2]} e^{n_j x_j} d(-P_j(x_j)) \tag{13}$$

$$= \prod_{j\in S} \left([-e^{n_j x_j} P_j(x_j)]_0^{\log(2)} + \int_{x_j\in[0,\log(2)]} n_j e^{n_j x_j} P_j(x_j) dx_j\right) \tag{14}$$

$$= \prod_{j\in S} \left(P_j(0) + \int_{x_j\in[0,\log(2)]} n_j e^{n_j x_j} P_j(x_j) dx_j\right)$$

$$\leq \prod_{j\in S} \left(e^{-n_j D_{\mathrm{KL}}(p_{j,a^*},1/2)} + \int_{x_j\in[0,\log(2)]} n_j e^{n_j x_j} e^{-n_j(x_j + C_1(p_{j,a^*},1/2))} dx_j\right) \tag{15}$$

$$= \prod_{j\in S} \left(e^{-n_j D_{\mathrm{KL}}(p_{j,a^*},1/2)} + \int_{x_j\in[0,\log(2)]} n_j e^{C_1(p_{j,a^*},1/2)} dx_j\right)$$

$$\leq \prod_{j\in S} \left(e^{-n_j D_{\mathrm{KL}}(p_{j,a^*},1/2)} + n_j e^{C_1(p_{j,a^*},1/2)}\right)$$

where (13) follows from the independence of the comparisons. We obtain (14) by applying integration by parts, (15) follows from the Chernoff bound and Fact E.1; here $C_1(\mu_1,\mu_2) = (\mu_1-\mu_2)^2/(2\mu_1(1-\mu_2))$, and the final inequality follows by observing that $\int_{x_j\in[0,\log(2)]} n_j e^{C_1(p_{j,a^*},1/2)} dx_j = n_j e^{C_1(p_{j,a^*},1/2)} \cdot \int_{x_j\in[0,\log(2)]} dx_j = n_j e^{C_1(p_{j,a^*},1/2)} \log(2)$. Note that log refers to the natural logarithm, so we have $\log(2) \leq 1$. Combined with (12), this completes the proof of (8). $\qquad\square$

### E.1.1 High-probability Regret Bound

We now prove Theorem E.1. Fix any $\delta > 0$. We first define event $E(\delta)$ as before.

**Definition E.1** (Event $E(\delta)$). *An estimate $\widehat{p}_{i,j}(r)$ in batch $r$ is **weakly-correct** if $|\widehat{p}_{i,j}(r) - p_{i,j}| \leq c_{i,j}(r)$. Let $C(\delta) := \lceil \frac{1}{2}\log_q(1/\delta)\rceil$. We say that event $E(\delta)$ occurs if for each batch $r \geq C(\delta)$, every estimate is weakly-correct.*

The next lemma shows that $E(\delta)$ occurs with probability at least $1-\delta$. Since $E(\delta)$ does not depend on the elimination criterion, its proof follows from the analysis of C2B.

**Lemma E.3.** *For all $\delta > 0$, we have*

$$\mathbf{P}(\neg E(\delta)) = \mathbf{P}\left(\exists r \geq C(\delta), i, j : |\widehat{p}_{i,j}(r) - p_{i,j}| > c_{i,j}(r)\right) \leq \delta.$$

As before, we analyze our algorithm under both events $G$ and $E(\delta)$. Recall that, under event $G$, the best arm $a^*$ is not eliminated. *Conditioned on these*, we next show:

- The best arm, $a^*$, is *not defeated* by any arm $i$ in any round $r > C(\delta)$ (Lemma E.4).
- Furthermore, there exists a round $r(\delta) \geq C(\delta)$ such that arm $a^*$ defeats *every other arm*, in every round after $r(\delta)$ (Lemma E.6).

We re-state the formal lemmas next.

**Lemma E.4.** *Conditioned on $G$ and $E(\delta)$, for any round $r > C(\delta)$, arm $a^*$ is not defeated by any other arm, i.e., $a^* \notin \cup_{i\neq a^*} D_r(i)$.*

To proceed, we need the following definitions.

**Definition E.2.** *The candidate $i_r$ of round $r$ is called the **champion** if $|D_r(i_r)| = |\mathcal{A}|-1$; that is, if $i_r$ defeats every other active arm.*

**Definition E.3.** *Let $r(\delta) \geq C(\delta) + 1$ be the smallest integer such that*

$$q^{r(\delta)} \geq 2A \log A, \qquad \text{where } A := \frac{32}{\Delta_{\min}^2} \cdot \log(2K^2).$$

We use the following inequality based on this choice of $r(\delta)$.

**Lemma E.5.** *The above choice of $r(\delta)$ satisfies*

$$q^r > \frac{8}{\Delta_{\min}^2} \cdot \log\left(2K^2 q_r\right), \qquad \forall r \geq r(\delta).$$

Then, we have the following.

**Lemma E.6.** *Conditioned on $G$ and $E(\delta)$, the best arm $a^*$ is the champion in every round $r > r(\delta)$.*

We are now ready to prove Theorem E.1.

*Proof of Theorem E.1.* First, recall that in round $r$ of C2B, any pair is compared $q_r = \lfloor q^r \rfloor$ times where $q = T^{1/B}$. Since $q^B = T$, C2B uses at most $B$ rounds.

For the rest of proof, we fix $\delta > 0$. We now analyze the regret incurred by C2B, conditioned on events $G$ and $E(\delta)$. Recall that $\mathbf{P}(G) \geq 1 - \frac{1}{T} \cdot e^{K \log(C) - f(K)}$ (Lemma E.2), and $\mathbf{P}(E(\delta)) \geq 1 - \delta$ (Lemma E.3). Thus, $\mathbf{P}(G \cap E(\delta)) \geq 1 - \delta - \frac{1}{T} \cdot e^{K \log(C) - f(K)}$. Let $R_1$ and $R_2$ denote the regret incurred before and after round $r(\delta)$ (see Definition E.3) respectively.

**Bounding $R_1$.** We can bound $R_1$ as in the proof of Theorem 1.1; so, we get

$$R_1 \leq O(K^2) \cdot \max\left\{ q \cdot \frac{\log K}{\Delta_{\min}^2} \cdot \log\left(\frac{\log K}{\Delta_{\min}}\right), \, q^2 \sqrt{\frac{1}{\delta}} \right\}. \tag{16}$$

**Bounding $R_2$.** This is the regret in rounds $r \geq r(\delta) + 1$. By Lemma E.6, arm $a^*$ is the champion in all these rounds. So, the only comparisons in these rounds are of the form $(a^*, j)$ for $j \in \mathcal{A}$.

Consider any arm $j \neq a^*$. Let $T_j$ be the total number of comparisons that $j$ participates in after round $r(\delta)$. Let $r$ be the penultimate round that $j$ is played in. We can assume that $r \geq r(\delta)$ (otherwise arm $j$ will never participate in rounds after $r(\delta)$, i.e., $T_j = 0$). As arm $j$ is *not* eliminated after round $r$,

$$I_j(r) - I^*(r) \leq \log(T) + f(K).$$

By Lemma E.6, $I^*(r) = 0$ (since $a^*$ is the *champion*, the summation is empty). So, we have $I_j(r) \leq \log(T) + f(K)$. Observe that

$$I_j(r) \geq D_{\mathrm{KL}}\left(\widehat{p}_{a^*,j}(r), \frac{1}{2}\right) N_{a^*,j}(r) \tag{17}$$

We can lower bound $D_{\mathrm{KL}}\left(\widehat{p}_{a^*,j}(r), \frac{1}{2}\right)$ as follows.

$$D_{\mathrm{KL}}\left(\widehat{p}_{a^*,j}(r), \frac{1}{2}\right) \geq \left(\widehat{p}_{a^*,j}(r) - \frac{1}{2}\right)^2 \geq \left(p_{a^*,j} - c_{a^*,j}(r) - \frac{1}{2}\right)^2 \geq \left(\frac{\Delta_j}{2}\right)^2$$

where the first inequality follows from Pinsker's inequality, the second inequality uses Lemma E.3 and the final inequality uses the fact that $c_{a^*,j}(r) \leq \frac{\Delta_{\min}}{2}$, which follows by the choice of $r(\delta)$. Plugging this into (17), we get

$$\frac{\Delta_j^2}{4} \cdot N_{a^*,j}(r) \leq \log(T) + f(K)$$

which on re-arranging gives

$$N_{a^*,j}(r) \leq \frac{4(\log(T) + f(K))}{\Delta_j^2}.$$

As $r + 1$ is the last round that $j$ is played in, and $j$ is only compared to $a^*$ in each round after $r(\delta)$,

$$T_j \; \leq \; N_{a^*,j}(r+1) \; \leq \; N_{a^*,j}(r) + 2q \cdot N_{a^*,j}(r) \; \leq \; \frac{12q \cdot (\log(T) + f(K))}{\Delta_j^2}.$$

The second inequality follows since $j$ is compared to $a^*$ in rounds $r$ and $r + 1$, and the number of comparisons in round $r + 1$ is $\lfloor q^{r+1} \rfloor \leq q \cdot (2q_r) \leq 2q \cdot N_{a^*,j}(r)$. Adding over all arms $j$, the total regret accumulated beyond round $r(\delta)$ is

$$R_2 = \sum_{j \neq a^*} T_j \Delta_j \leq \sum_{j \neq a^*} O\left( \frac{q \cdot (\log(T) + f(K))}{\Delta_j} \right). \tag{18}$$

Combining (16) and (18), and using $q = T^{1/B}$, we obtain

$$R(T) \leq O\left( T^{1/B} \cdot \frac{K^2 \log(K)}{\Delta_{\min}^2} \cdot \log\left( \frac{\log K}{\Delta_{\min}} \right) \right) \;\; + O\left( T^{2/B} \cdot K^2 \cdot \sqrt{\frac{1}{\delta}} \right) + \sum_{j \neq a^*} O\left( \frac{T^{1/B} \cdot \log(T)}{\Delta_j} \right)$$

$$+ \sum_{j \neq a^*} O\left( \frac{T^{1/B} \cdot f(K)}{\Delta_j} \right)$$

This completes the proof Theorem E.1. $\qquad\square$

## F Hardware Specification for Computational Experiments

We conducted our computations using C++ and Python 2.7 with a 2.3 Ghz Intel Core $i5$ processor and 16 GB 2133 MHz LPDDR3 memory.