# OpenReview forum: "An Asymptotically Optimal Batched Algorithm for the Dueling Bandit Problem"
_NeurIPS.cc/2022/Conference — NeurIPS 2022 Accept_

### Official Review · Reviewer_iZS2 · 2022-07-08

**Rating:** 6
**Confidence:** 4
**Soundness:** 3 good
**Presentation:** 3 good
**Contribution:** 3 good

**Summary:**

This paper studies the dueling bandit problem with regret minimization as the objective under the assumption of an existing Condorcet winner. In contrast to most existing work dealing with fully adaptive learning situations, this work considers the situation of batched learning, where the learner does not decide before each learning round which pairwise comparison to perform, but decides this only in a limited number of rounds, i.e., it performs a batch of the same pairwise comparison at certain learning rounds. For this learning scenario, a learning algorithm is proposed that has a near-optimal upper bound on its cumulative regret using a nearly optimal number of batches. This improves upon existing methods which additionally need stronger assumptions than merely the existence of a Condorcet winner. The suggested algorithm is compared in a numerical study on real-world data sets with existing regret minimizing learning algorithms for the dueling bandit setting.

**Questions:**

Although the main ideas of the algorithm are described quite well, I would have liked to see a remark on the usage of two different confidence intervals in Section 3. I was wondering if this is technically not even one of the key design ideas which improve the theoretical results?

Moreover, the textual description in Section 3 could be connected to the high-level description in Section 1.1 by reusing the term “trapped” and “recourse”, as it does not appear in Section 3 anymore.

Finally, I am wondering why the theoretical analysis is not carried out for the KL-divergence-based confidence intervals if it (a) leads to the same theoretical results and (b) performs better empirically?

### Minor things:

- Typo in line 165: von Neumann.
- There is a typo in [35].
- In the math display of Proof of Lemma 3.1. it should be $\hat p_{i,j}(r)$
- The idea of the proof of Theorem 1.2 goes back to the proof of RUCB’s expected regret bound. I think the RUCB authors deserve some more credit.


**Limitations:**

## Potential negative social impact
As this is a theoretical work, there is no seemingly potential negative social impact.

## Limitations
The authors clearly state that their algorithm is based on the Condorcet winner assumption and mention that it would be desirable to consider learning settings without such an assumption, e.g. Copeland winner or von Neumann winner settings.


**Strengths And Weaknesses:**

# Strengths

- Batched learning settings are definitely relevant for practical applications and have also been studied recently in the standard multi-armed bandit setting. Thus, the scope of the paper is relevant to the ML community.

- The paper is generally well-written. The notation is well thought out and the main ideas underlying the suggested learning algorithm are described in sufficient detail.
- Although the learning setting is not novel, as it has been considered by other authors before, the suggested learning algorithm comes with stronger theoretical guarantees than existing methods.
- The proofs seem to be sound as far as I can tell.

# Weaknesses

## Related literature

There is some missing related literature:

-  Mark Braverman, Jieming Mao, and S. Matthew Weinberg. Parallel algorithms for select and partition with noisy comparisons. In Proceedings of the ACM symposium on Theory of Computing, pages 851- 862, 2016.

This paper also studies the question of round/batch complexity of learners in the dueling bandit setting, but for the objective of finding the top-k arms.

- Chuang-Chieh Lin and Chi-Jen Lu. Efficient mechanisms for peer grading and dueling bandits. In Asian Conference on Machine Learning (ACML), pages 740-755, 2018.

This paper studies the question of round/batch complexity of learners in the multi-dueling bandit setting (with utilities of the arms), but for the objective of finding the best arm (NB: the Borda winner scenario is considered as well).

Although the objectives of the two papers are different from the considered regret minimization objective, I think they are definitely worth mentioning.

As a side note: These two papers are discussed in the following survey paper on dueling bandits (Section 3.1.18, 3.2.6, and 4.2.8):

- Bengs, V., Busa-Fekete, R., El Mesaoudi-Paul, A., & Hüllermeier, E. (2021). Preference-based Online Learning with Dueling Bandits: A Survey. Journal of Machine Learning Research, 22(7), 1-108.

Moreover, there are some missing references (lines 166-167) regarding the multiple arm comparison for the regret minimization objective (see Chapter 6 of Bengs et al. (2021)).

## Experimental study

The experimental study leaves room for improvement. First, the considered competing algorithms BTM and RUCB are out of date and Double Thompson Sampling and Self-Sparring are currently the state-of-art (together with the considered RMED). Next, it is a bit doubtful that the elimination strategy is changed for the experiments, although the theoretical analysis uses a Hoeffding-based elimination strategy. While I agree with the author’s statement that *the theoretical guarantee will continue to hold*, it is nevertheless something that needs to be shown. Thus, the authors should include the original algorithm in the experiments and call the one using KL elimination differently in the experiments. Finally, it would be nice if the authors could comment on whether the data sets satisfy SST or STI, i.e., the assumptions needed for SCOMP and SCOMP2, and also report the number of batches used by C2B and SCOMP2.

NB: There is also a quite recent Python package for dueling bandits: https://duelpy.gitlab.io/duelpy/index.html

---

> ### Author Response · Authors · 2022-08-02
> **Re: KL-divergence based proof, related work, experiments**
>
> Thank you for your comments and feedback. We answer the main questions raised in the review.
>
> $Re: \ KL-divergence \ based \ proof$. For completeness, we now include a complete theoretical analysis of our algorithm with a KL-based elimination criterion (see rebuttal supplemental, Appendix E).
> We remark that the proof is straightforward and follows along the lines of Komiyama et al. (2015).
> We chose to present the Hoeffding-based analysis as it is cleaner; specifically, we can easily show that the best arm is never deleted (conditioned on a good event).
> Lastly, it is well-known that KL-based criterion is better in practice than Hoeffding-based criterion (Komiyama et al. (2015)), and it is a standard practice to use KL-based criterion in experiments.
>
> $Re: \ Using \ different \ confidence \ intervals \ for \ defeat \ and \ elimination$.  Indeed, the confidence intervals are carefully designed to make the analysis clean. The $\gamma_{i,j}$ confidence interval (which is typically larger) is designed to ensure that the best arm is never eliminated. The $c_{i,j}$ confidence interval is crucially used in $trapping$ the best arm so we can use it as an anchor; also note that this confidence interval only kicks in after $r(\delta)$ rounds.
>
>
> $Re:\ Related \ work.$ Thanks for bringing these references to our attention. We will include these references in Section 2. Furthermore, since there has been extensive work on the dueling bandit problem, we will also add references to relevant surveys.
>
> $Re: \ Experiments.$ Our goal was to investigate the batched setting for the dueling bandit problem under the existence of a Condorcet winner, and try to achieve tight asymptotic regret guarantees using a small number of batches (which we managed to do). For the computational study, we considered a representative set of sequential algorithms as benchmarks: our goal here is to demonstrate that we can achieve competitive performance against the sequential algorithms using a small number of batches. We chose RUCB and RMED for comparison since our algorithm is based on a similar paradigm, and comparing against those would allow us to study the $power \ of \ adaptivity$ in this context.
>
> Thanks for pointing us towards the recent python package for dueling bandits - using it, we will include a comparison against double Thompson sampling (DTS) and self-sparring in the final version of the paper. Also note that for the Condorcet setting, the regret curves of RMED and DTS are similar (it has been observed in prior work of Wu and Liu (2016)); so, we expect that the central message of our paper would remain unchanged.
>
> Most real world datasets (that we are aware of) do not satisfy the SST+STI assumptions; nor do the ones we use.
> Lastly, we use the same number of batches for both SCOMP2 and C2B ($\log(T)$ and $\log(T)+6$).
> We will clarify these points in the experimental section.
>
> $Re: \ Minor \ comments.$ We will fix the typos. We will add a note under Section 1.4 comparing our algorithm to RUCB in the final version.

---

> > ### Comment · Reviewer_iZS2 · 2022-08-07
> > **Re: Re: KL-divergence based proof, related work, experiments**
> >
> > Dear authors,
> >
> > many thanks for your response!
> >
> > ### Re: KL-divergence proof.
> > Great! I agree that it is straightforward to use a KL-based elimination criteria and show theoretical results. Therefore, I have no doubts that you have done there a proper job.
> >
> > ### Re: Using different confidence intervals for defeat and elimination.
> > You should mention this in the corresponding section to improve readability.
> >
> > ### Re:Experiments.
> > Now, I see why you have chosen RUCB and RMED. Thanks for the clarification! Perhaps you should mention this in the final version, i.e.
> > *We chose RUCB and RMED for comparison since our algorithm is based on a similar paradigm, and comparing against those would allow us to study the power of adaptivity in this context.*

---

### Official Review · Reviewer_AugK · 2022-07-11

**Rating:** 6
**Confidence:** 4
**Soundness:** 3 good
**Presentation:** 3 good
**Contribution:** 3 good

**Summary:**

This work considered the batched dueling bandit problem under the Condorcet condition. It proposed a sequential elimination-based algorithm and derived regret bounds in expectation and with high probability. It also presented numerical experiments to show the effectiveness of the proposed algorithm.

**Questions:**

1. Are the authors willing to provide a regret lower bound? If it matches the derived lower bound, it would be more convincing that the upper bound is tight and the algorithm is optimal. There should be some discussions in Section 1.4.
2. As the algorithms are proposed for different settings, how are they compared? It is nice that the algorithms are compared using several datasets, but more explanation of the experiment setup is appreciated.
3. In Section 2, in addition to the detailed discussion, the readers may get a even clear view if a table comparing the bounds and conditions of algorithms in the existing literature is presented.

**Limitations:**

There seems to be no societal issue from my view.

**Strengths And Weaknesses:**

Strengths: The paper is well organized and is easy to follow. It clearly discussed the motivation of problem setting, algorithm design and the technique challenges in detail and evaluate the algorithm with real-life datasets.

Weaknesses: A lower bound may help strengthen the theoretical results and more explanation in the experiments setup is appreciated. (See "Questions" part for more details.)

---

> ### Author Response · Authors · 2022-08-02
> **Re: Lower bound, experiments and presentation**
>
> Thank you for your comments and feedback. We answer the main questions raised in the review.
>
> $Re: \ Lower \  bound$. Note that our batched algorithm obtains $tight \ asymptotic \ regret$; that is, when $\lim_{T \to \infty}\frac{R_T}{\log(T)} = O\left(K/\Delta_{\min}\right)$. For finite $T$, the batched lower bound of $\Omega\left(T^{1/B}/{\Delta_{\min}}\right)$ and the sequential lower bound of $O\left(\frac{K}{\Delta_{\min}}\text{poly}(\log(T))\right)$ from prior work applies. There was an additive gap of $O(K^2/{\Delta_{\min}^2})$. However, recently Saha and Gaillard (2022) gave the first $sequential$ algorithm that obtains $O\left(\frac{K}{\Delta_{\min}}\text{poly}(\log(T))\right)$ regret.
> As stated in our response to Reviewer Zi8f, this result does not imply any improvements for the batched dueling bandits problem in a straightforward manner. We leave it as future work to close this additive gap.
>
> $Re: \ Experiments$. Although the algorithms we compare are designed for different settings (batched v/s sequential), we compare the cumulative regret incurred by the algorithms. Specifically, we allow the sequential algorithms to work as prescribed; that is, they work in $B = T$ batches. For the batched algorithms (SCOMP2 and C2B), we study two settings: (i) $B=\log(T)$, and (ii)$B=\log(T)$+6. Our results further corroborate the fact that ``small'' number of batches suffice to achieve good regret guarantees (against the more powerful sequential algorithms).
> We will clarify our setup in the computational section in the final version.
>
>  $Re: \ Presentation$. This is a good suggestion. We will add a table in the final  version to  compare the regret guarantees obtained by the various algorithms and their corresponding setting (sequential v/s batched).

---

> > ### Comment · Reviewer_AugK · 2022-08-08
> > **Rebuttal is acknowledged**
> >
> > Thanks the authors for clarification. I increased the rating.

---

### Official Review · Reviewer_Zi8f · 2022-07-16

**Rating:** 6
**Confidence:** 3
**Soundness:** 3 good
**Presentation:** 3 good
**Contribution:** 3 good

**Summary:**

The authors study the batched dueling bandit problem under the Condorcet condition. They design an algorithm named C2B with a few adaptive rounds, and derive the upper bound for the asymptotic regret. Computational experiments are carried out with a number of real-world datasets.

**Questions:**

1. The authors claim that their method can nearly achieve the best rate for regret (line 12 and 115). Meanwhile, they mention a lower bound of $O(\frac{K}{\Delta_{\min}}\cdot poly(\log T))$ (line 113), while the regret bound they derive is $\tilde{O}(\frac{K^2}{\Delta_{\min}^2})$. It seems a gap exists here, especially when $K$ is large or $\Delta_{\min}$. Is it possible to diminish the gap by trade-off the number of batches with performance?

2. In the proposed algorithm, whenever a pair $(i,j)$ is compared in round $r$, it is always compared $q_r = \lfloor q^r\rfloor$ times. Will it be useful to adaptively assign the number of times for compare at each round based on previous compare results, in order to improve the performance? Intuitively, we might want to compare a pair more times if it is harder to tell which one is better.

**Limitations:**

The authors discuss the limitations of their work to some extent. There seems no negative societal impact.

**Strengths And Weaknesses:**

Strength:
1. The authors show that the proposed algorithm can achieve good performance with number of batches as small as $O(\log t)$, which seems an interesting and new result.
2. The feature of few adaptive rounds of the algorithm might accommodate modern applications of large scale.
3. Overall, the paper is well-organized and easy to follow. The paper is detailed in backgrounds as well.

Weakness:
Please find my questions/concerns in the next part.

---

> ### Author Response · Authors · 2022-08-02
> **Re: Lower bound and #comparisons in each batch**
>
> Thank you for your comments and feedback. We answer the main questions raised in the review.
>
> (1) $Re: \ Lower \ bound$. Note that our batched algorithm obtains $tight \ asymptotic \ regret$; that is, when $\lim_{T \to \infty}\frac{R_T}{\log(T)} = O\left(K/\Delta_{\min}\right)$ for $B = O(\log T)$.
> We note that even popular sequential algorithms like RMED and RUCB suffer $O\left(\frac{K \log(T)}{\Delta_{\min}}+ \frac{K^2}{\Delta_{\min}^2}\right)$ regret similar to our regret upper bound with $B = O(\log T)$. Increasing $B$ be beyond $O(\log T)$ does not improve our regret upper bound; improving the dependence on $K$ is a subject for future work.
>
>
>
> $Re:\ Comparisons \ in \ each \ batch$. The reason for having geometrically increasing number of comparisons is to exactly capture the intuition that pairs that are similar may require a greater number of comparisons before we are able to declare a winner (as you observe). However, since there is no apriori information, we must start with $T^{1/B}$ and build from there. When a arm survives round $r$, all we can determine is that $T^{-r/2B}$ was not the correct gap; we cannot rule out $T^{-(r+1)/2B}$.
> In our algorithm, we choose the number of comparisons (per round) in a simple way: equal number for each compared  pair. This already provides a good regret bound, which is   asymptotically optimal.
> Designing policies that adaptively determine the number of comparisons for each pair based on the instance is interesting, and a good direction for future research.

---

### Author Response · Authors · 2022-08-02
**Rebuttal Revision: Theoretical Analysis of KL-divergence based Elimination Criterion (Appendix E)**

We now include a complete theoretical analysis of the KL-divergence based elimination criterion in Appendix E of the supplemental file.

---

### Meta-Review · Area_Chair_sP81 · 2022-08-30

**Recommendation:** Accept
**Confidence:** Less certain

**Metareview:**

This paper makes nice progress in the "dueling bandits" framework by giving a near-optimal regret bound under (exponentially) fewer rounds of adaptivity.  This makes theoretical progress on this problem and may have real-world applications in settings where adaptivity is costly or difficult to implement with each round.

**Award:**

No

---

### Meta-Review · Area_Chair_jQiC · 2022-09-08

**Recommendation:** Reject
**Confidence:** Certain

**Metareview:**

This paper studies a batched dueling bandit problem under the Condorcet condition. The authors propose a sequential elimination-based algorithm and derive regret guarantees. Numerical experiments are reported at the end. The paper improves upon [1], where the problem has been introduced. The reviewers praised the paper but did not give strong support to it. I went over the paper myself. The results are of some interest, but indeed not utterly compelling/surprising. In general, one is left wondering what is the main technical difficulty in batching a stochastic dueling bandit algorithm over and beyond batching a standard (non-dueling) bandit one.
Putting everything together, and considering the very high level of competition in this year's Neurips conference, I regret to say that this paper does not reach the acceptance threshold.

**Award:**

No

---

### Decision · Program_Chairs · 2022-09-14

Accept